# Discerning Developmental Dyscalculia and Neurodevelopmental Models of Numerical Cognition in a Disadvantaged Educational Context

**DOI:** 10.3390/brainsci12050653

**Published:** 2022-05-16

**Authors:** Flavia H. Santos, Fabiana S. Ribeiro, Ana Luiza Dias-Piovezana, Caterina Primi, Ann Dowker, Michael von Aster

**Affiliations:** 1Affective, Behavioural and Cognitive Neuroscience, School of Psychology, University College Dublin, D04 V1W8 Dublin, Ireland; 2Department of Social Sciences, Faculty of Humanities, Education and Social Sciences, University of Luxembourg, L-4366 Esch-Sur-Alzette, Luxembourg; fabiana.ribeiro@uni.lu; 3Faculty of Sciences, São Paulo State University, Bauru 17033-360, Brazil; analuiza_dias@yahoo.com.br; 4Department of Neuroscience, University of Florence, 50139 Florence, Italy; caterina.primi@unifi.it; 5Department of Experimental Psychology, University of Oxford, Oxford OX2 6GG, UK; ann.dowker@psy.ox.ac.uk; 6Department of Psychology, University of Potsdam, 14469 Potsdam, Germany; m.aster@drk-kliniken-berlin.de; 7Children’s Research Center, University Children’s Hospital Zürich, 8032 Zurich, Switzerland

**Keywords:** developmental dyscalculia, prevalence, neuropsychology

## Abstract

Developmental Dyscalculia (DD) signifies a failure in representing quantities, which impairs the performance of basic math operations and schooling achievement during childhood. The lack of specificity in assessment measures and respective cut-offs are the most challenging factors to identify children with DD, particularly in disadvantaged educational contexts. This research is focused on a numerical cognition battery for children, designed to diagnose DD through 12 subtests. The aims of the present study were twofold: to examine the prevalence of DD in a country with generally low educational attainment, by comparing z-scores and percentiles, and to test three neurodevelopmental models of numerical cognition based on performance in this battery. Participants were 304 Brazilian school children aged 7–12 years of both sexes (143 girls), assessed by the Zareki-R. Performances on subtests and the total score increase with age without gender differences. The prevalence of DD was 4.6% using the fifth percentile and increased to 7.4% via z-score (in total 22 out of 304 children were diagnosed with DD). We suggest that a minus 1.5 standard deviation in the total score of the Zareki-R is a useful criterion in the clinical or educational context. Nevertheless, a percentile ≤ 5 seems more suitable for research purposes, especially in developing countries because the socioeconomic environment or/and educational background are strong confounder factors to diagnosis. The four-factor structure, based on von Aster and Shalev’s model of numerical cognition (Number Sense, Number Comprehension, Number Production and Calculation), was the best model, with significant correlations ranging from 0.89 to 0.97 at the 0.001 level.

## 1. Introduction

Numerical cognition can be defined as the ability to represent quantities in both cognitive and neural systems, through innate and acquired numerical skills [1]. In general, there is an age-related mastering of numerical cognition aptitudes, which relies on the development of other cognitive abilities (such as language, working memory, spatial abilities, executive functions, etc.) and their respective neural substrates, as well as being influenced by formal education [2]. Essentially, neurodevelopmental dysfunctions of numerical cognition may be defined as “transient and mild” or “persistent and severe” [3]; the former is referred to as “low achievement in math”, and although it impacts school learning, it is more responsive to treatment or may spontaneously disappear; the latter is often considered as a specific learning disorder, termed Developmental Dyscalculia (DD; [4]) or Mathematical Learning Disorder, whose deficits are already observed at preschool age [2,5,6,7,8]. The present study is focused on a numerical cognition battery assessment in children targeting the identification of the prevalence of DD in school children immersed in a generally disadvantaged educational environment and testing its dimensionality considering two models of numerical cognition.

### 1.1. Challenges for Prevalence Studies

According to the International Consensus, DD is defined as “a heterogeneous disorder that produces individual differences in both development and functioning of numerical cognition, evidence-based in neuroanatomical, neuropsychological, and behavioural levels, as well as their interactions” [9] (pp. 1–3). The current classification of primary or secondary DD is based on aetiological elements. Primary DD comprises specific severe numeracy deficits, with no other complications; it is relatively rare and has a prevalence from 1 to 2% in school children [5,10,11]. On the other hand, secondary DD accounts for around 4% of the cases [5] whose numerical dysfunctions are accompanied by equally severe “non-numerical” cognitive deficits relative to chronological age or schooling [3,12], for instance, a recent study observed that attentional deficits were a core cognitive marker of secondary DD [13]. In addition, secondary DD includes comorbidity with other neurodevelopmental disorders, for instance, dyslexia or attention deficit hyperactivity disorder ICD-11 [14].

There are no universally accepted criteria for the diagnosis of DD. The currently general criteria acknowledged include: (i) a discrepancy with intelligence measures; (ii) cut-off scores on standardized measures of numerical cognition; (iii) inconsistency in years of schooling (delay); (iv) resistance to interventions [15]. Different diagnostic manuals adopt slightly dissimilar general criteria, and there is considerable debate in the scientific community about which psychometric criteria are the most appropriate [16]. For example, some studies use a cut-off point between the 20th and 35th percentile on standardized tests as indicative of numeracy deficits [6,7,15,17], while others use a stricter cut-off point of scores below the 10th or even 5th percentile [7,18,19]. Obviously, the more lenient the cut-off point, the higher the prevalence will be. Intelligence discrepancy is a controversial criterion, first by conceptual definition [16]. It varies across studies, both in terms of the size of the discrepancy required, and in terms of whether they include or exclude children with average or near-average mathematics scores but extremely high intellectual quotient (IQ) [19,20,21]. DSM-V [16,22] and ICD-11 [14] concur that a specific learning disorder diverges from general learning difficulties associated with intellectual disability, both recognise below-average IQ as a confounding factor, along with congenital encephalopathy [23] and very-preterm birth [24]. However, despite not meeting the criteria for DD, children with lower IQ scores or brain injuries will also need support to learn math. For an updated overview of the diagnostic criteria see Castaldi, Piazza, and Iuculano [25]. Most epidemiological studies were carried out in developed countries and have suggested a prevalence of DD between 3% and 6.5% [5,15,26]. However, especially given the lack of agreed diagnostic criteria, studies indicating a higher prevalence than the average (and even some that do not) may be grouping together remarkably diverse categories of mathematical difficulties [27]. These studies may include both children who have intrinsic and severe difficulties with numerical concepts and children who are low attainers in mathematics due to social (e.g., poverty, late start in school, poor attendance, lack of books, low parental education, etc.) or educational factors, for instance, poor-performing schools [28]. This problem is a major concern, especially in countries with low overall educational attainment [29] where education is not standardized and there are many disadvantaged schools [30].

Therefore, studying the prevalence of a specific learning disorder, in a developing country such as Brazil, requires several variables to be considered. For instance, the Basic Education Development Index indicates that the quality of schools and resources for education are not equivalent in all Brazilian regions [31]; consequently, low school achievement by international standards, such as the PISA study [32], is in common and accentuates the gender gap, especially in mathematics [30,33]. For instance, there are three Brazilian epidemiological studies of DD. Ribeiro and Santos [34] used a two-phase diagnostic technique, involving screening followed by neuropsychological assessment, with a cohort of 407 students aged 8 or 9 years, enrolled in the 3rd school year of four public schools in the countryside of São Paulo State, and found 22 (5.4%) of the children to have DD. Fortes et al. [35] carried out a cross-sectional study of 1618 students from the 2nd to 6th grades in four regions in Brazil, using DSM-5 criteria for dyscalculia, a school achievement test and controlling for the variables of age, city, socioeconomic status, gender and IQ, and found a prevalence of 6.0%. Bastos et al. [36], using a mathematical screening test, found a higher prevalence (7.8%) in a cohort of 2893 (N = 128) with a greater frequency of boys.

It is still uncertain whether gender-related differences in mathematics performance depend more on school grade or age because these variables are usually overlapping [37]. Genetic distance measures do not seem to be a major determinant of gender differences in mathematics [38], especially given the fact that these disparities have reduced significantly in more gender-equal societies. In developed countries, such gender gaps in mathematics performance have declined progressively [39]. Environmental factors that usually shape individual differences in mathematics (e.g., characteristics of parents, socioeconomic status and schools) do not contribute to high or low achievement in mathematics nor to boys’ and girls’ differences in performance [40]. However, social roles and social expectations modulate a child’s behaviour in all spheres, especially in academic ones, for instance, mothers and teachers tend to underestimate girls’ mathematics performance compared to boys [15,41], and this may elicit a long-lasting negative impact on the recruitment and retention of women in science, technology, engineering, and mathematics in adult life.

Finally, prevalence conclusions are also constrained by the variety of means of assessment for dyscalculia. School achievement and screening measures are useful for identifying low attainers in numeracy. However, these measures have sensitivity but lack specificity to identify dyscalculia since there are many environmental causes for low attainment in mathematics [18,29,30,33]. Moreover, a number of cognitive tasks involving domain-general and domain-specific measures show low diagnostic power and accuracy in school children [42,43]. Therefore, low attainers in numeracy benefit from further neuropsychological testing for diagnosis purposes. In comparison, numerical cognition batteries are designed to test for specific deficits and establish the diagnosis of dyscalculia when appropriate. The battery used in this study, Zareki-R [44], may be seen as a potential advance in the study of DD and its diagnosis. It consists of a wide variety of subtests, explored in the next section, measuring different components of number processing and calculation. Moreover, it has already been translated successfully into several languages and is used in many countries such as Switzerland, Germany, France, Belgium, Brazil, Algeria, etc. [45,46,47,48,49,50]. Therefore, this numerical cognition battery [5] shows promise for cross-cultural studies (e.g., [48]). For instance, Santos et al. [49] assessed 172 Brazilian children, aged 7–12 years from public schools in urban and rural areas. The study found high to moderate correlations between the subtests of this battery and the Arithmetic subtest of WISC-III, indicating good construct validity (r < 0.65). As expected, younger children obtained a lower global score than older children. Regarding rural children, the teaching method had a greater effect on performance than the home environment. Boys outperformed girls in 3 out of 12 tasks (Mental calculation, Problem-solving and Oral comparison); however, the gender effect size was small for the Mental calculation and Oral comparison subtests and medium for the Problem-solving subtest.

### 1.2. Neurodevelopmental Models of Numerical Cognition

To test the dimensionality of the Zareki-R, we selected two theoretical models previously studied concerning the battery. These models are complementary rather than antagonists. Stanislas Dehaene proposed the Triple Code Model [51], which postulates that the architecture of number processing is composed of three systems: analogue magnitude or number sense (the endowed ability to estimate small quantities in a set, observed in several species), verbal (vocabulary, auditory and spoken knowledge related to quantities and numerals) and visual (the symbolic representation of quantities and numerals) codes. The more versed a person is in dealing with quantities and numerals, the stronger becomes the relationship between the three codes, which allows transcoding, that is, an automatic transfer from one code to another. Schooling intervenes as the core contributor to the development of an internal metric of quantities, the mental number line, which deals with large quantities and precise calculation [52]. The model is supported by studies carried out with infants, children, adults and monkeys [53]. The numerical cognition model of von Aster and Shalev [2], expands Dehaene’s triple code by adding a fourth component, the ordinal system, which appears later in childhood as mathematical reasoning itself. Thus, during childhood, numerical cognition jolts on the cardinal system or the approximate number system, which perceives small quantities without the need for counting. Factors such as age, life experiences and education progressively support the development of verbal (words related to quantities, such as small/big, more/less, first/second, etc.), symbolic (e.g., the Arabic numerals in modern cultures) and then ordinal systems [10,49]. The development of the four systems occurs gradually and in parallel with other cognitive functions, particularly working memory. von Aster and Shalev’s model is supported by cognitive, neuropsychological and neuroimaging studies, connecting arithmetic and working memory brain networks [2,3,29,45]. The transition from number sense to a mental number line in Dehaene’s model theoretically corresponds to the transition from the cardinal to the ordinal systems in von Aster and Shalev’s model.

Zhang et al. [54] investigated the dimensionality of a preliminary version of the battery (known as NUCALC in English) in a sample of 310 Chinese schoolchildren based on the Triple Code Model [51]. The battery subtests were divided into three modules reciprocally connected: *Analogue Magnitude* (positions on a vertical scale; oral comparison; perceptual estimation; contextual estimation), *Verbal code* (counting dots; counting backwards, mental calculation; memory for digits, problem-solving) and *Visual Arabic* (dictation of numbers; reading numbers; written comparison). From a developmental perspective, the analogue code is the inherent capacity to establish relationships between a given magnitude to a set of items, while the other codes are acquired through experience and formal education [55]. These three codes are expected to have independent trajectories but overlap based on children’s age, schooling and experiences allowing the transcoding automatization [56]. Neuroimaging studies, as summarised by [57,58,59,60] have described brain circuits that form the neural substrate underlying neurodevelopmental trajectories of these codes, including children with DD. However, Zhang et al. [54] observed that the developmental trajectories of these codes from grade 1 to grade 4 are not identical; the visual Arabic increases across the four grades, while the other two modules achieve a plateau at the third grade, perhaps because children are exposed to them before schooling.

The dimensionality of the Zareki-R was tested with subjects in preschool and 2nd grade in a follow-up study of 307 Swiss children [5]. A four-factor solution was found at preschool age using an equivalent kindergarten battery, the Zareki-K (K stands for kindergarten). The factors were Arabic notation, visual analogic, subitizing/estimation and working memory, which were, respectively, responsible for 35.9%, 8.7%, 6.9% and 6.4% of the observed variance. For 2nd graders, a three-factor solution was found for Zareki-R, i.e., Arabic notation, evaluation of quantities and counting. These factors explained, respectively, 37.8%, 8.6% and 7.0% of the observed variance [5]. However, these factors were not testing a specific theoretical model, and, in some cases, partial scores were used rather than total scores. In another report, Santos et al. [61] organised the Zareki-R subtests into four constructs based on the total score of each subtest: *Number Sense*, composed of the sum of Counting dots and Perceptual estimation scores; *Number Comprehension*, composed by the sum of Oral comparison, Written comparison and Contextual estimation scores; *Number Production*, composed by the sum of Counting backwards, Dictation of numbers and Reading numbers scores; *Calculation*, composed by the sum of Mental Calculation, Problem-solving scores and Positioning numbers. The performance of children from 1st to 6th grade was age-related but not gender-related among constructs, except for *Number Comprehension*; *Number Production and Calculation* composites were correlated with working memory (r < 0.36; *p* < 0.001), corroborating behavioural and neuroimaging studies including typically developing children and children with developmental dyscalculia [2,3,45,57,59]. Nevertheless, the authors did not test the dimensionality of those constructs psychometrically in depth. In the present study, we test to what extent the battery results correspond to the neurodevelopmental model of numerical cognition proposed by von Aster and Shalev [2] of four factors (Number Sense, Number Comprehension, Number Production and Calculation) versus Dehaene’s Triple Code Model including analogue magnitude, verbal and visual Arabic codes [51,55,58,60]. A possibility that has not been tested yet is the higher-order solution, a factor analysis that allows testing of the hierarchical structure of the model, this approach could answer whether von Aster and Shalev’s model has a core mathematical cognition (MC) factor.

The aims of the present study were twofold. First, we aimed to estimate the prevalence of DD in a developing country contrasting z-scores and percentiles in numerical cognition tasks. Considering the clinical relevance, we performed two methods to evaluate and standardize test performance: the z-scores and percentiles to estimate the prevalence of DD. The use of the z-score is recommended by the World Health Organization [62] since it reflects the reference population distribution as standardized measures; z-scores are comparable across age, sex and measure (as a measure of “dimensionless quantity”). Nevertheless, a limitation of z-scores is that they are not straightforward to explain to the public and may be of limited use in clinical settings. On the other side, the percentile is related to the position of a subject in a given reference distribution. Percentiles are easier to understand and to use in practice, both by health professionals and the public since they dictate the expected percentage of a population should be above (or below) a given score [63]. Second, we sought to investigate the theoretical neurodevelopmental model of numerical cognition [2], based on a battery for the diagnosis of DD. In order to conduct meaningful multigroup comparisons, it is necessary to show that the measurement instrument is operating equally in the compared groups [64,65,66]. Specifically, three models were tested, the von Aster and Shalev four-factor structure [2], the higher-order mathematical cognition solution and Dehaene’s triple code structure [51]. Additionally, we tested the gender invariance of the higher-order model across genders.

## 2. Materials and Methods

### 2.1. Participants

Participants were 304 children (161 boys) aged 7–12 years enrolled from 1st to 7th grades. The age groups and school grades were equivalent in all cases, meaning no cases of scholar delay or grade repetition. In regard to gender, the children were distributed across age bands: fifteen boys of age 7, forty-seven boys of age 8, thirty-five boys of age 9, thirty-two boys of age 10, fourteen boys of age 11 and eighteen boys aged 12. The children were recruited from government schools sited in five urban areas in Southeast Brazil, precisely Assis, Ourinhos, Bauru, São José do Rio Preto and São Paulo cities. The schools were selected according to two criteria: (a) being public (State) schools; (b) including children in the target age bands.

The inclusion criterion was an IQ within the normal range according to CID-11 [14]. Since participants were recruited for two independent projects described previously [49], different measures were used: 169 children in the sample were assessed by the Raven’s Coloured Progressive Matrices (inclusion criterion was scoring between the 24th and 75th percentile, *M* = 67.08, *SD* = 21.00 [67], and 134 children were assessed by the Wechsler Intelligence Scale for Children—WISC-III (inclusion criterion was an IQ score between 80 and 120, *M* = 105.47, *SD* = 12.19 [68]. For gender contrasts per instrument, see Appendix A.

All children were Brazilian nationals and native monolingual Portuguese speakers. According to parent and teacher reports, none of the participants had known specific learning disorders, emotional disturbances, motor deficits, speech, or hearing impairments, or neurological or psychiatric diagnoses.

### 2.2. Screening and Domain-Specific Measures

Raven’s Coloured Progressive Matrices (Brazilian adaptation, [67]). A measure of abstract non-verbal reasoning that was designed as a measure of general cognitive ability. It is composed of three series, each with 12 matrices: A, Ab and B. The matrices are arranged in increasing order of difficulty within each series, each series being more difficult than the previous one. The items consist of a drawing or matrix with a missing part. Below the main drawing, six alternatives are presented, one which correctly completes the array. The child must choose one of the alternatives that correspond to the missing part.

Wechsler Intelligence Scale for Children (WISC) 3rd Edition (Brazilian adaptation, [68]) is an individually administered intelligence test for children between the ages of 6 and 16. In the current study, verbal IQ was calculated using the subtests Vocabulary, Similarities, Arithmetic and Digit Span: (i) Vocabulary: The child is asked to define aloud a given word. After six consecutive errors, the task is discontinued. (ii) Arithmetic: The child has to solve orally presented arithmetic story problems. After three consecutive errors this task is discontinued; (iii) Similarities: Two words are presented in each item (e.g., “wood and coal”), and the child is asked in which way they are similar. The task is discontinued after 8 consecutive errors; (iv) Digit Span: Children are given sequences of numbers orally and asked to repeat them, as heard and in reverse order.

School Achievement Test (SAT): This test comprises three subtests: Writing, Visual Arithmetic and Reading. [69]. In this study, the arithmetic subtest was used to assess oral and written calculations. Each item of this subtest presents a range of calculations in ascending order of difficulty, which are presented to children of all school grades.

Zareki-R—Battery of Neuropsychological Tests for Number Processing and Calculation in Children—Revised (Brazilian adaptation, [49]) is an international specialized pencil-and-paper battery test that assesses numerical cognition in school-age children. Composed of 12 subtests: (i) *Counting dots*—Children must enumerate different sets of dots. (ii) *Counting backwards*. The participant must count the dots backwards, e.g., from 23 to 1 and from 67 to 54; (iii) *Dictation of numbers*. The child is asked to write, in Arabic numerals, eight orally presented numbers (e.g., [23]); (iv) *Mental calculation*, in which eight additions, eight subtractions and six multiplications are presented orally; (v) *Reading Numbers*: The participant must read eight numbers written in Arabic numerals, such as 15 and 1900; (vi) *Positioning numbers*: In this subtest a vertical number line is presented, in which the participant is asked to point and mark a specific position said by the experimenter; (vii) *Oral comparison*: Eight pairs of numbers are verbally presented (e.g., 34,601 and 9678) and the child must judge which one is the largest in quantity; (viii) *Perceptual estimation*: The child must give an oral estimate of the quantity of items shown in a picture, which is displayed for 5 s (e.g., 57 balls); (ix) *Contextual estimation*: The child must judge sentences with regard to the size of quantities in a context, for instance, whether “eight lamps in the same room” is “little”, “medium” or a “lot”?; (x) *Problem solving*: The child must solve orally presented numerical word problems of increasing difficulty. For instance, one of the problems is, “Peter has 12 marbles. He gives 5 to his friend Ann. How many marbles does Peter have now?”; (xi) *Written comparison*: Pairs of numbers in Arabic numeral form are presented visually, for example, 13 and 31, and the child must judge which one is the largest of the pair; (xii) The *Memory for Digits* is a working memory measure that requires the forward (FDS) and backward (BDS) repetition of digit sequences of increasing length. The battery is administered in full to all participants, items may receive 0, 1 or 2 points depending on the subtest or the quality of the response, being 0 for incorrect and 2 for accurate and without cues or repetitions. The total score is the sum of all subtests except memory for digits [70]. Additionally, under Dellatolas et al. [48], Score A concerning schooling achievement was calculated by adding the scores of the following six subtests of Zareki-R: Dictation of numbers, Reading numbers, Mental calculation, Problem-solving, Oral comparison and Written comparison. Zareki-R total score and memory for digits were dependent variables analysed separately.

### 2.3. Procedures

Written consent was obtained from the participating schools and the parents/guardians of the children prior to testing. It was explained to each child that the experiment could be discontinued at any time. The study was conducted following the Declaration of Helsinki and approved by the Ethics Committee of UNESP, São Paulo State University for studies involving humans (protocol code 0095/2005). Parents also filled out an interview form (adapted from [71]) about the child’s medical, social, educational and psychological development. Both the child participants and their parents received information about the aims and procedures, also about the freedom to discontinue the activities anytime without any impact on their studies or grades. Children who consented were assessed individually in their own schools in a quiet room. Screening measures (schooling achievement test and intellectual level) were assessed in a previous neuropsychological session. Zareki-R was administered in a single 30-min (on average) session; the order of subtests was not fixed, and verbal tasks were alternated with nonverbal ones to avoid fatigue.

### 2.4. Data Analysis

Concerning the statistical analyses, univariate distributions of each subtest were examined for assessment of normality, considering Skewness and Kurtosis indices of the items (ranges outside the values of −1 and 1 indicate non-acceptable departures from normality).

As a first step, the four-factor structure, the higher four-factor solution and Dehaene’s triple code structure were tested by confirmatory factor analysis (CFA) employing the mean-adjusted maximum likelihood (MLM) estimator (Mplus software; [72]). This estimator provides the Satorra–Bentler Scaled chi-square (SBχ^2^; [73]), a adjusted and robust measure of fit for non-normal sample data, which is more accurate than the ordinary chi-square statistic [74]. To test the models’ fit, the following indices were considered: the ratio of chi-square to its degrees of freedom (SBχ^2^/df), the comparative fit index (CFI; [75]), the Tucker–Lewis Index (TLI; [76]), and the root mean square error of approximation (RMSEA; [77]). In the case of χ^2^/df, values below or equal to two are considered good, while values between two and three are considered acceptable [78]. For the TLI and CFI indices, values above 0.90 indicate acceptable fit, while values above 0.95 indicate excellent fit [79]. The RMSEA value is considered acceptable when it is below 0.08 and good when it is below 0.05 [80]. Furthermore, we used the Akaike Information Criterion (AIC; [81] and the Bayesian Information Criterion (BIC; [82]) to compare the different models and to choose the model that presented the lowest level of loss of information. Concerning the AIC and BIC indices, the model that minimizes those indices can be selected as the best model (see [83], for a discussion about AIC and BIC indices).

Then, gender invariance analyses were conducted by performing hierarchically nested CFAs, and gender invariance was evaluated using not only Δχ^2^, which is sensitive to sample size, but also ΔCFI, which has been found to be the most sensitive index to detect a lack of invariance [84], employing the absolute value of ΔCFI of less than 0.01 [64,85].

We also carried out a multivariate analysis of covariance (MANCOVA), including the Zareki-R subtests as dependent variables, excluding only Memory for Digits as described in the battery handbook, having gender (boys versus girls) as the independent variable and ages as a covariate variable. Multiple comparisons were controlled by the Bonferroni test. Finally, the present study also provides empirical support for the scale reliability and validity as described in the results section since it has been underexplored.

## 3. Results

### 3.1. Prevalence Criteria

Criterion 1—Percentile obtained from the total score. When percentiles were calculated based on the Zareki-R total score, 14 children obtained low scores, as defined by a total score below the 5th percentile. According to this criterion, the prevalence of DD was 4.6%. Table 1 and Table A1 present the results obtained by percentile through the total score and subtests, respectively.

Criterion 2—The Zareki-R subtests and total scores were converted into *z*-scores (see Table A1) by subtracting the score from the total sample mean at the first assessment and dividing the difference by the standard deviation (SD). Twenty-two participants from the total cohort (the same 14 plus 8 additional children) performed 1.5 standard deviations below average in the Zareki-R total [5,45]. According to IQ screening measures, the discrepancy between performance and intelligence was sustained in all cases. According to this criterion, the prevalence of DD in the present sample was 7.4%.

In Appendix Table A2 the individual performance of the 22 children with DD revealed that two tasks were the most affected in both younger and older children: counting backwards and problem-solving. One-third of the children with DD failed in these tasks, i.e., achieved only 0, 1 or 2 row score points. Extremely low performance was also observed in Perceptual estimation for six children from 7 to 10 years. Figure 1 indicates two opposite patterns in the performance of children with DD. In some tasks, errors increased with age, meaning that performance worsens as a function of task complexity: mental calculation, reading numbers, memory of digits, context estimation and problem-solving. On the other tasks, errors decreased with age, that is, children can master some abilities across grades such as: counting dots, counting backwards, dictate numbers, positioning numbers, oral comparison, perceptual estimation and written comparison.

### 3.2. Dimensionality

From von Aster and Shalev’s framework, we tested two models, one with only four correlated factors (i.e., *Number Sense, Number Comprehension*; *Number Production and Calculation*) and one hierarchical with four factors loading on a higher-order factor where the covariation between the four factors was accounted for by a higher-order mathematical cognition (MC) factor.

Preliminarily, the four-factor structure (*Number Sense, Number Comprehension*, *Number Production and Calculation)* was tested by confirmatory factor analysis (CFA). Results showed that goodness of fit indices for the four-factor model were all adequate (SBχ^2^/df = 2.2; CF = 0.97; TLI= 0.96; RMSEA = 0.06; AIC= 140.543; BIC= 244.620), Figure 2. Then the higher-order mathematical cognition solution (i.e., the four factors plus a *mathematics cognition* quotient (MC)) were tested by confirmatory factor analysis (CFA). Results showed that goodness of fit indices for the higher-order mathematical cognition solution were all adequate (SBχ^2^/df = 2.2; CFI= 0.97; TLI= 0.96; RMSEA= 0.06; AIC= 139.215; BIC= 235.858), Figure 3. Finally, the Dehaene’s triple code structure (*Analogical Magnitude*, *Verbal code and Visual Arabic*) was tested. Results showed the goodness of fit of the model (SBχ^2^/df = 1.7; CFI= 0.98; TLI= 0.97; RMSEA= 0.05; AIC= 143.440; BIC= 243.800), Figure 4.

Comparing the three models, the higher four-factor solution model had lower values for the Information Criterion indices (AIC and BIC) than the other models and for this reason, it can be considered the best model. In the higher-order mathematical cognition solution, standardized factor loadings ranged from 0.30 to 0.92, and were significant at the 0.001 level. The correlations between the four factors and the higher-order mathematical cognition solution were all significant (from 0.89 to 0.97).

### 3.3. Gender Invariance

As a prerequisite, we tested the final higher-order model separately per gender [61]. The model showed acceptable or good fit indices among boys (*χ*^2^/*df* = 2.33; CFI = 0.94; TLI = 0.91; RMSEA = 0.08) and for girls (*χ*^2^/*df* = 1.43; CFI = 0.98; TLI = 0.96; RMSEA = 0.06).

To test gender invariance, in line with the recommended practice for testing measurement invariance [64,65,86], first the independence model was fitted (*χ*^2^ = 1476.79, *df* = 90, *p* < 0.001). As reported in Table 2, in addition to configural invariance, the first-order factor loadings were equal across genders. Then, scalar, or strict invariance, which constrained intercepts to be invariant across groups, and, subsequently, the equivalence of the second-order factor loadings, was supported. Finally, after having tested those structural variances and covariances were invariant across gender, the equality of the items’ variances and covariances was confirmed. We also detected a lack of invariance employing the absolute value of ΔCFI that was less than 0.01 by the more restrictive model.

### 3.4. Gender Differences

Having preliminarily verified the measurement equivalence of the scale, we carried out a multivariate analysis of covariance (MANCOVA), including all the Zareki-R subtests as dependent variables, excluding only Memory for Digits, having gender (boys versus girls) as the independent variable and ages as a covariate variable. Results showed no differences between genders after controlling for age, *F* (11, 291) = 1.46, *p* = 0.14; *Wilk’s* Λ = 0.95, *η_p_*^2^ = 0.05. Furthermore, Memory for Digits, Zareki-R Total, and Score A were analysed as dependent variables through separated ANCOVAs, with gender as the independent variable and age as a covariate. Outcomes revealed no significant main effects of gender for Memory for Digits; (*F* (1301) = 0.005; *p* = 0.94; *η_p_*^2^ < 0.001) and for Zareki-R Total (*F* (1301) = 3.61, *p* = 0.06; *η_p_*^2^ = 0.01). However, there was a borderline significant effect of gender on Score A (*F* (1301) = 4.03; *p* = 0.05; *η_p_*^2^ = 0.01), with boys performing better than girls.

In order to investigate the neurodevelopmental model of numerical cognition, we carried out a multivariate analysis of variance (MANOVA) having as dependent variables the Zareki-R subtests, except for Memory for Digits, and ages as independent variables. Wilks’ test showed a significant age effect on the Zareki-R subtests; *F*(55, 1337) = 5.93, *p* < 0.001; *Wilk’s* Λ = 0.36, *η_p_*^2^ = 0.18. Moreover, Tukey post hoc tests were used with a significant alpha level of P ≤ 0.05. Results are presented in Table 3.

### 3.5. Reliability

To assess test-retest reliability on the Zareki-R scores, 14 typically developing children, with a mean age of 8.71 years (SD 0.61) performed the numerical cognition battery twice, with a 63.14 days (SD 11.47) interval between the two testing sessions. The correlations between pre- and post-test for all Zareki-R subtests are shown as Appendix A.

A Wilcoxon Signed-Ranks Test showed that the median post-test scores were statistically significantly higher than the median pre-test scores for the following tasks: Raven’s Coloured Progressive Matrices, Counting Recall, Memory of Digits and Zareki-R Total, as would be predicted from the children’s increased age and school experience.

### 3.6. Criterion-Related Validity

A Pearson product-moment correlation coefficient was computed to assess the relationship between the Zareki-R subtests, the Arithmetic subtest of WISC-III and the Arithmetic subtest of SAT. Results showed a positive moderate correlation between the Arithmetic subtest of WISC-III with six Zareki-R subtests (Counting backwards, Dictation of numbers, Mental calculation, Reading numbers, Oral comparison, Problem-solving) and positive strong correlations with the Zareki-R Total Score and Score A. Moreover, SAT arithmetic subtest correlations were positive and moderate for five subtests of Zareki-R (Counting backwards, Dictation of numbers, Reading numbers, Oral comparison, Contextual estimation), and were positive and strongly related to two Zareki-R subtests (Mental calculation and Problem solving), to Zareki-R Total Score and Score A. For results, see Appendix A. Overall, the external validity of Zareki-R subtests, total and Score A was confirmed as the scores of the arithmetic subtests of the WISC-III and the SAT correlated significantly.

## 4. Discussion

The present cross-sectional study aimed (i) to contrast three theoretical neurodevelopmental models of numerical cognition based on a battery for diagnosis of DD, respectively, the four-factor structure [2,10], the higher four-factor solution and Dehaene’s triple code structure [51], (ii) to estimate the prevalence of DD in a country with generally low education attainment, comparing z-scores and percentiles in numerical cognition tasks. For this purpose, we obtained age- and gender-related normative data in a sample of 304 Brazilian school children, we also supplied further psychometric information, such as invariance, external and internal validity, and reliability.

As expected, an age-related effect was observed, corroborating previous national [10,49] and international [45,47,50] studies. Ceiling effects were observed in six subtests for some age bands (Counting dots, Counting backwards, Dictation of numbers, Reading numbers, Mental calculation and Positioning numbers), indicating that these tests represent achieved competencies, while the other five subtests (Written comparison, Problem-solving, Contextual estimation, Perceptual estimation and Oral comparison) were more challenging for the typically developing participants. In a cohort of typically developing Brazilians, it was observed that the four systems of numerical cognition are rudimentarily functional even in preschool [87], which explains why some abilities at primary school can achieve ceiling effects. Our findings also corroborate the trajectory study of Zhang [54] in the sense that only the visual code continues to progress, while the verbal and analogical codes achieve a plateau by the 3rd grade. Eventually, a future review of the battery could add more complex items. Children with DD presented deficits in these tasks, in some cases resulting in floor effects (Table A2). As with Santos et al. [61], no main effect of gender was found throughout the subtests, probably because the present sample is more representative than in the preliminary study [49], including five cities. In terms of gender parity and equality, the rate of participation in primary education is similar for Brazilian boys and girls [88]. Moreover, the school attendance in this age band is higher than 98% nationwide, and we believe that this might enable greater equality in mathematics performance between boys and girls [15,89]. Apart from that, all participants were from coeducational schools, which are the commonest type of school in Brazil.

Concerning the nature of both mathematical development and deficits, this study supports the view that numerical cognition is not a single entity, but is multifaceted in varied cohorts, including typically developing children [89,90,91,92] as well as children with low numeracy [93,94,95]. Although, Zareki-R was originally designed considering Dehaene’s Triple Code [51,55,58], testing its dimensionality based on the performance of our sample revealed that the four-factor model was the best model, meaning that subtests are tackling interdependent components. At least during primary school, the four numerical cognition systems [2], i.e., cardinal, verbal, symbolic and ordinal (respectively, *Number Sense, Number Comprehension*, *Number Production and Calculation*) were highly and significantly correlated components, in which children may show selective strengths and weaknesses. A caveat needed is that subtests are designed to test specific numerical cognition abilities, and as with cognitive tests from other areas, tasks are multisensorial in nature, so some overlap between these systems is inevitable. The results suggest that the Zareki-R tests have utility both for the diagnosis of DD, and for the understanding of different arithmetic components and the relationships and discrepancies between them, both in typical and atypical mathematical development. However, this study did not aim to answer the question of whether there are specific foundational numerical abilities that are invariably impaired in children with DD. Apart from the severity marker, we consistently found deficits in at least three subtests for children under percentile 5: problem-solving, counting backwards and mental calculation. We also observed that in those children identified as having DD, some abilities might worsen with age, indicating a different trajectory [6]. The test battery used here, as with similar test batteries, is likely to be a suitable resource to test hypotheses about such potential foundational abilities. A close inspection of the cases of DD detected by the battery strongly underlines the individual differences [96]. Based on the total score of the numerical cognition battery, we obtained two different measures to estimate DD prevalence. Fourteen children (7 boys) met the performance criterion below the 5th percentile, while 22 children (11 boys) met the z-score criterion of −1.5 SD from the mean total score. Both rates, respectively, 4.6% and 7.4%, are within the average prevalence range for dyscalculia [5,15,34,35]. The instrument also allows, for clinical usage, the criterion of deficits below 1.5 *SD* in at least three subtests, which inflates the rate (27 cases, 8.88%) and resembles less restrictive studies [6,7,15,17,36]. We consider that the two criteria, 1.5 SD in the total score or three subtests, are clinically useful for the rapid identification of those who need intervention, although lower scores in three subtests only should be used with caution in environments with generally low attainment. However, to avoid ambiguity in research carried out in developing countries [31,32,33], we recommend a stricter criterion that is the 5th percentile, which leads to a prevalence comparable to the global average [5,15,26]. the criterion adopted can be used as a marker to determine the intensity of the intervention required.

More generally speaking, the adopted criterion may work as a marker to determine the intensity of the proposed intervention. Children scoring below the 5th percentile may require very intensive intervention, while those with higher scores may benefit from lighter touch interventions [97,98]. Thus, to meet the criteria from medical manuals, we do recommend that the diagnosis of DD using a test battery should be complemented by other sources and resources, keeping in mind the nature and strictness of the cut-off adopted [7,8,9]. Moreover, in clinical settings within developing countries, a second assessment after six months is highly recommended to confirm the persistence or worsening of difficulties [19,99], especially to disentangle the effects of socioeconomic and educational disadvantage, which are confounding factors for specific learning disorders.

Limitations of the present study include the small number of children aged 11 and 12 years, as well as the fact that the sample was regional rather than national, although international studies used similar sample sizes [2,45,47,50]. Further research is needed concerning the generalizability of the findings. Notably, it would be desirable to investigate whether the structure of the components of early numerical cognition unveiled in the present study is similar in different countries and education systems and whether the same factors predict dyscalculia in different environments.

Studies using a test battery such as the Zareki-R may help to cast light on both the extent to which DD should be seen as a severe form of low mathematical attainment versus a distinct entity and the extent to which it should be seen as homogeneous versus heterogeneous. The present study suggests that dyscalculia is distinguished from low mathematical attainment, given that it can be found even within a context where educational limitations resulting in low mathematical attainment are common. Neuroimaging studies that used the Zareki-R for diagnosis, showed that children with DD have reduced grey and white matter volumes in brain areas related to numeracy, therefore, the severity of symptoms is combined with differences in brain activation [45,57,59]. Moreover, performance at preschool age is predictive of DD [2,5,6]. Our study also confirms that DD is clinically an extremely heterogeneous condition (see cases in Appendix Table A2); it is indeed possible for individuals to show marked discrepancies between almost any two potential numerical cognition tasks.

This research targeted numerical cognition components controlling for key diagnosis criteria, such as age and grade discrepancy, among other controls. However, there are several different potential causes for mathematical deficits in the general population, e.g., difficulties in core number skills; visual-spatial abilities; language; reasoning; memory capacity [100]. Consequently, interventions may vary according to the determinant factors, nature and severity of the deficits [101]. Assuming the argument that there is a single entity that can be called dyscalculia, this term should be restricted to those problems that are caused by a specific deficit in core number skills (e.g., [4,102]), preferentially examined through operationalised diagnosis. Further studies, especially cross-national and cross-cultural studies, may help to elucidate the relationships between core numerical abilities and performance on different components of numeracy, independent of the effects of specific teaching methods and curricula.

The Zareki-R and similar batteries may also prove useful in testing the effects of educational interventions (e.g., [34,103,104]) both for individuals with dyscalculia diagnoses and for those with low mathematical attainment caused by other factors such as educational disadvantage. They will make it easier to evaluate the overall effects of interventions and to investigate whether given components are particularly susceptible to certain interventions. Thus, they will enable both a greater theoretical understanding of the relationships between, and influences on, different components of numeracy and a greater practical understanding of how to plan and test effective interventions for children with mathematical difficulties.

## 5. Conclusions

In the present study, we tested the dimensionality of three theoretical neurodevelopmental models of numerical cognition based on a battery for the diagnosis of DD and estimated the prevalence of DD by comparing z-scores and percentiles per subtests and total score. Complementarily we presented further psychometric properties of the battery. The study venue was Brazil, a developing country that has consistently performed below the OECD average and without evidence of progression across all editions of the PISA study [32]. In a sample of 304 scholar children aged 7 to 12 years old enrolled in mixed public schools, we observed age-related but no gender-related differences. Although all models showed goodness of fit, the four-factor model based on Von Aster and Shalev [2] was the best model. Concerning prevalence, the stricter criterion, i.e., the 5th percentile, has proven to be ideal for research, while the z-score criterion of −1.5 SD from the mean seems ideal for clinical purposes, especially considering intervention. The percentile criterion detected 14 children and the z-score added eight cases. Both rates, respectively, 4.6% and 7.4%, are within the average prevalence range for DD and were balanced by gender.

## Figures and Tables

**Figure 1 brainsci-12-00653-f001:**
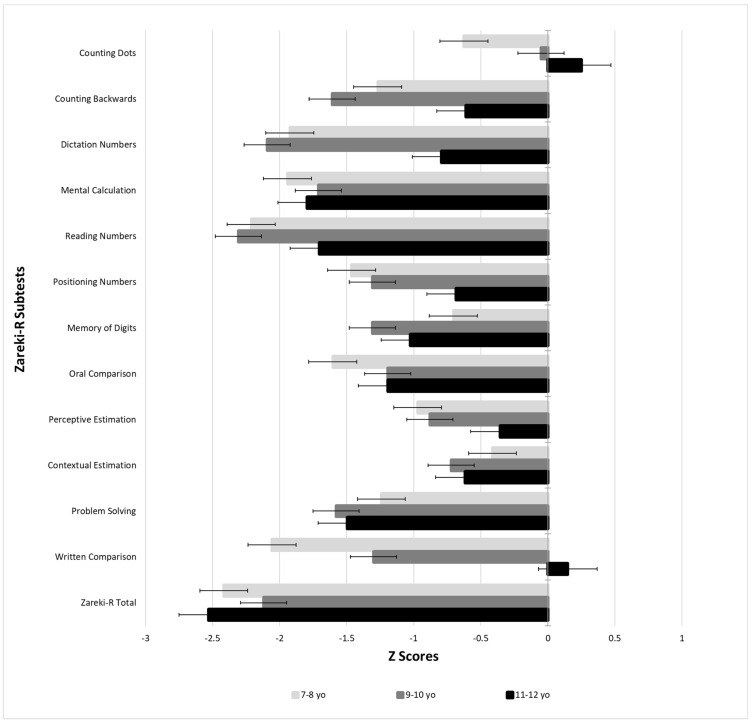
Performance on Zareki-R of the children with DD (N = 22), per age band.

**Figure 2 brainsci-12-00653-f002:**
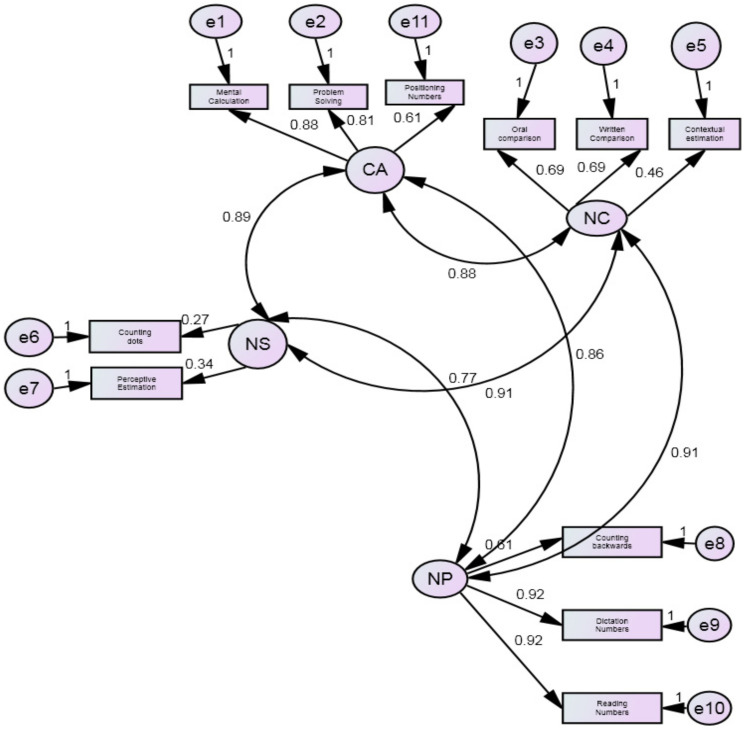
The four factors structure: CA= Calculation, NS= Number Sense, NC= Number Calculation, NP= Number Production.

**Figure 3 brainsci-12-00653-f003:**
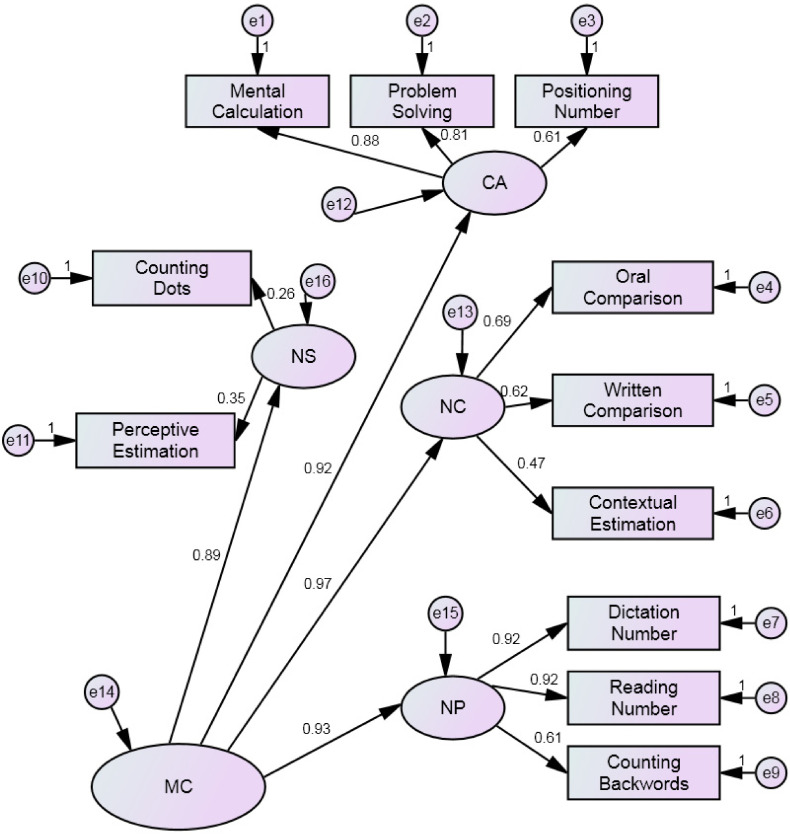
The higher-order mathematical cognition solution (i.e., the four factors plus a mathematics cognition quotient (MC)).

**Figure 4 brainsci-12-00653-f004:**
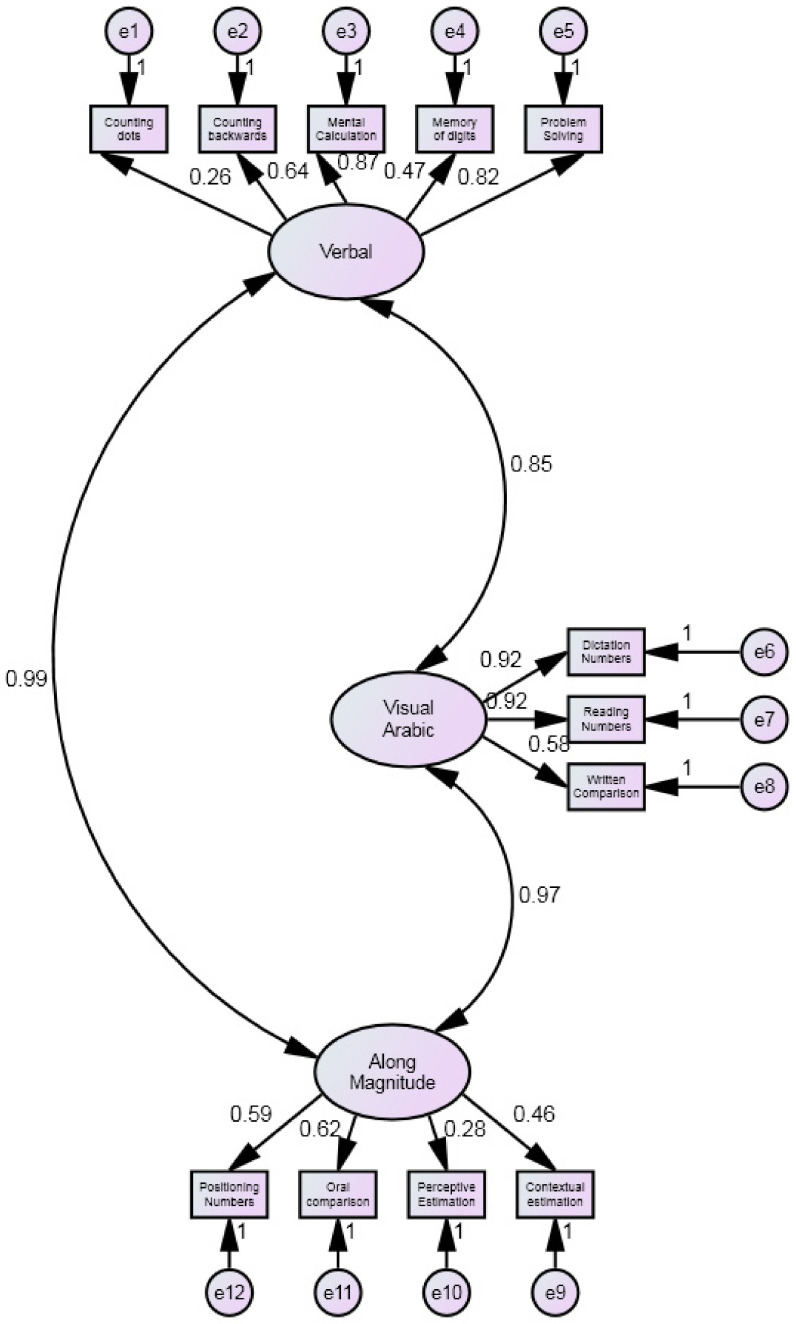
Dehaene’s triple code structure (*Analogical Magnitude*, *Verbal code and Visual Arabic*).

**Table 1 brainsci-12-00653-t001:** Range of raw score results for Zareki-R total, number of students for each per percentile and classification by age groups.

Age 7	N (36)	Age 8	N (81)	Age 9	N (75)	Age 10	N (51)	Age 11	N (28)	Age 12	N (33)	Percentile	Classification
>138.95	1	>160	3	>174.8	4	>176.1	3	>173	2	>174.35	1	**>95**	High
104.2–135.0	8	144.2–159.8	17	158.0–174.2	15	165.0–175.4	10	168.1–173.0	5	165.2–172.9	9	**75–94**	High average
65.9–103.1	18	111.6–143.8	40	127.8–157.2	37	149.8–164.7	25	148.1–167.7	14	143.9–165.1	15	**26–74**	Average
48.2–64.6	8	75.5–111.2	17	85.1–127	15	125.0–149.5	10	119.6–147.5	6	131.2–143.75	7	**6–25**	Low average
<45.4	1	<75.0	4	<84	4	<122.7	3	<114.1	1	<123.6	1	**<5**	Low

**Table 2 brainsci-12-00653-t002:** Goodness of fit statistics for each level of structural and measurement invariance across genders.

Model	*χ*^2^ (*df*)	CFI	Model Comparison	Δ*χ*^2^	Δ*df*	*p*	ΔCFI
1. Invariance of model configuration	116.22 (62)	0.961	-	-	-	-	-
2. Invariance of first-order factor loadings	125.34 (68)	0.959	Model 1–Model 2	9.12	6	0.167	0.002
3. Invariance of intercepts	141.78 (78)	0.954	Model 2–Model 3	16.44	10	0.088	0.005
4. Invariance of second-order factor loadings	144.36 (81)	0.954	Model 3–Model 4	2.58	3	0.462	0.000
5. Invariance of structural variances/covariances	150.11 (86)	0.954	Model 4–Model 5	5.75	5	0.331	0.000
6. Invariance of measurement error variances/covariances	159.78 (96)	0.954	Model 5–Model 6	9.67	10	0.470	0.000

Note: *χ*^2^ = chi-square test; *df* = degrees of freedom; CFI = robust comparative fit index; Δ*χ*^2^ = Satorra–Bentler scaled difference; Δ*df* = difference in degrees of freedom between nested models; *p* = probability value of Δ*χ*^2^ test; ΔCFI = difference between robust CFIs of nested models.

**Table 3 brainsci-12-00653-t003:** Performance on Zareki-R subtests in Brazilian children per age and gender.

	Girls (*n* = 143)	Boys (*n* = 161)	Age 7 (*n* = 36)	Age 8 (*n* = 81)	Age 9 (*n* = 75)	Age 10 (*n*= 51)	Age 11 (*n* = 28)	Age 12 (*n*= 33)	F _(5398)_	*p*	*η_p_* ^2^
Counting dots	3.44 (0.77)	3.42 (0.83)	3.36 (0.93)	3.21 (0.86)	3.37 (0.78)	3.61 (0.69)	3.68 (0.67)	3.70 (0.64)	3.24	0.007	0.05
Counting backwards ^a^	2.99 (1.25)	3.08 (1.18)	2.03 (1.36)	2.84 (1.34)	3.05 (1.21)	3.57 (0.73)	3.50 (0.88)	3.36 (0.82)	9.88	<0.001	0.14
Dictation of numbers ^b^	12.59 (4.29)	13.42 (3.74)	6.33 (3.90)	12.48 (3.60)	13.66 (3.66)	15.28 (1.12)	15.21 (1.26)	14.94 (1.39)	45.40	<0.001	0.44
Mental calculation ^c^	26.54 (11.12)	28.40 (11.65)	12.19 (9.00)	23. 95 (10.06)	29.11 (10.26)	35.26 (6.52)	34.68 (8.85)	31.46 (7.75)	35.16	<0.001	0.37
Reading numbers ^d^	13.66 (3.77)	14.32 (3.45)	7.92 (4.63)	13.72 (3.22)	14.65 (2.84)	15.78 (0.67)	15.93 (0.38)	15.55 (1.06)	45.72	<0.001	0.43
Memory of Digits	23.44 (6.21)	23.56 (6.57)	21.34 (5.64)	23.63 (6.72)	22.59 (5.71)	24.90 (6.90)	23.86 (6.88)	25.21 (6.02)	2.16	<0.06	0.03
Positioning numbers^e^	16.04 (5.06)	16.61 (4.85)	11.13 (5.88)	15.93 (4.67)	16.81 (5.25)	18.28 (2.91)	17.95 (3.82)	17.65 (3.01)	13.00	<0.001	0.18
Oral comparison ^f^	13.23 (2.84)	14.06 (2.45)	10.92 (3.42)	12.52 (2.88)	14.52 (1.83)	14.57 (1.66)	15.04 (1.23)	15.00 (1.50)	22.24	<0.001	0.27
Perceptual estimation	6.10 (2.27)	6.71 (2.29)	5.50 (2.36)	6.05 (2.36)	6.56 (2.41)	6.90 (2.05)	6.54 (1.90)	7.21 (2.13)	2.97	0.01	0.05
Contextual estimation ^g^	11.80 (4.89)	11.90 (5.11)	8.39 (3.99)	10.05 (4.53)	11.39 (5.00)	14.12 (4.81)	14.21 (3.86)	15.64 (3.41)	15.98	<0.001	0.21
Problem-solving ^h^	6.41 (3.91)	7.56 (3.92)	2.81 (3.25)	5.30 (3.57)	7.81 (3.97)	8.92 (2.63)	9.14 (2.86)	9.30 (2.47)	25.32	<0.001	0.30
Written comparison ^j^	18.73 (2.12)	18.88 (2.14)	16.72 (3.03)	18.57 (2.49)	19.05 (1.32)	19.49 (1.59)	19.57 (0.84)	19.39 (1.37)	11.21	<0.001	0.16
Zareki-R Total ^c^	140.27 (38.52)	149.20 (37.68)	87.31 (26.11)	124.75 (25.46)	140.00 (26.8)	155.78 (13.87)	155.54 (15.9)	153.26 (15.17)	53.47	<0.001	0.47
Score A ^c^	91.16 (23.31)	96.63 (23.46)	56.89 (20.38)	86.53 (19.59)	98.80 (19.67)	109.29 (10.42)	109.57 (12.22)	105.64 (11.83)	52.97	<0.001	0.47

Age effect by MANOVA for subtests and Age effect by MANCOVA, gender as covariant for ZAREKI-R Total; Tukey post-hoc: (a) 7 < 8–12 and 8 < 10; (b) 7 < 8–12, 8 < 10–12, and 9 < 10; (c) 7 < 8–12, 8 < 9–12, and 9 < 10; (d) 7 < 8–12 and 8 < 10–12; (e) 7 < 8–12, 8 < 10, and 9 < 10; (f) 7 < 8–12 and 8 < 9–12; (g) 7 < 9–12, 8 < 10–12, and 9 < 10–12; (h) 7 < 8–12 and 8 < 9–12; 9 < 10–11; (j) 7 < 8–12. *p* ≤ 0.05 in all cases. ZAREKI-R = Neuropsychological Tests Battery of for Number Processing and Mental Calculation in children, revised; N= number of participants; M= mean; SD= standard deviation; Score A= is calculated by the sum of the six following subtests of ZAREKI-R: dictation of number, reading numbers, mental calculation, problem solving, oral comparison, and written comparison.

## Data Availability

Data set will be made available via the OSF profile of the first author. Link: osf.io/963uz. (accessed on 3 May 2022).

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
