# Peer review of "Discerning Developmental Dyscalculia and Neurodevelopmental Models of Numerical Cognition in a Disadvantaged Educational Context"

_brainsci, 2022, doi:10.3390/brainsci12050653_

Round 1

Reviewer 1 Report

Journal: Brain Sciences

Reference number: 1594211

Authors: Santos, Ribeiro, Dias-Piovezana, Primi, Dowker and von Aster

Many thanks for the opportunity to review this manuscript. This manuscript is addressing an interesting question, but I have major concerns about the argument for this. The introduction lacks detail regarding the theoretical background for the models that this manuscript is contrasting. Specific comments relating to this can be found in the “introduction” subsection below.

Below I have provided comments relating to each section of the manuscript, followed by a subsection with minor points (e.g., typos etc.).

Abstract

  1. I recommend re-reading the abstract as some of the language use is unclear. This includes “accomplishment of elementary”, “based on a battery [of what?]” and “scholar children”. Doing this will enhance clarity of the abstract.
  2. Specify the low educational attainment country
  3. Be careful with the use of the term “both genders” as society has broadened its understanding of gender and now encompasses terms such as non-binary.
  4. What were the four factors in the best-fit model? This is needed to understand the significant correlations mentioned also.
  5. Can you add a few words to explain what the 1.5 z score is considered useful for?
  6. Three models are mentioned here (line 20) but then the introduction talks only about two (von Aster and Shalev vs. triple code model).

Introduction

The major limitation of the introduction is that it does not adequately explain the models of numerical cognition, the skills that form these models and how they are assessed, and what has been found in relation to both these models in previous literature.

  1. In the first paragraph (line 34), it could be useful to include parentheses offering examples of the “other cognitive abilities”
  2. On line 43, is this meant to refer to International Classification of Disease rather than International Consensus?
  3. What is the frequency of secondary DD – this is needed so the reader more fully understands the difference in prevalence rather between primary and secondary forms.
  4. Point three could be better explained (line 56). Do you mean inconsistency between expected school year for chronological age and actual school year? Or is it an inconsistency between expected math performance for their school year and observed math performance?
  5. What is considered “high prevalence” (see line 75)?
  6. More information is needed on the difference between low attainers in mathematics and those with severe difficulties given that this is a distinction that is important for this study.
  7. For readers outside of Brazil, it would be beneficial to clarify the age of students in year 3 (see line 89). This could also be clarified for all other year groups mentioned.
  8. Sentence starting on line 101 and continuing to 104 is unclear. This could be re-worded
  9. Supporting citations for environmental factors have the stronger effect are needed.
  10. Much more information is needed regarding the numeracy related skills that are critical to the development of mathematics. As it stands, after reading the introduction the reader has only a limited understanding of what these skills are.
  11. Paragraph starting on line 127. In the first line you need to clarify who test you are referring to as this is a new paragraph.
  12. Line 135 to 136: consider writing “as summarised by 45-48” rather than “such as a systematic review…”
  13. Line 149: observed variance in what?
  14. What is important about NC, NP and CA being related to phonological memory?
  15. As you are aiming to compare two models of numerical cognition (von Aster and Shalev vs. triple code model) both of these need to be explained in a lot more detail, along with the measures used to assess the aspects of these models. How do they overlap? What has previous research found in relation to each of these models? How do these models relate to your justification for conducting this research in Brazil?
  16. What compared groups are you referring to here (line 171)? More information is needed.
  17. What is the four-factor solution, the higher-four factor solution? These terms have not been used before until the final paragraph of the introduction.
  18. The introduction does not provide sufficient detail to support why you are contrasting z scores and percentiles.

Method

  1. Raven’s coloured progressive matrices assesses non-verbal cognition while the WISC assesses full-scale IQ. These measures are not directly comparable, nor are simply comparing the non-verbal component of the WISC to the RCPM given the differences in the assessments. Additional analysis needs to be conducted to see whether there is a difference in your key findings depending on whether the children met the inclusion criteria using the RCPM or WISC. This is critical as otherwise you are assuming that average intelligence on a non-verbal measure is the same as average intelligence on a more comprehensive measure.
  2. Dictation of numbers: how many digits were in each number? Was there a range (e.g., single digits to triple digits)? More information needing.
  3. Was there a discontinuation of any of the Zareki-R assessments? Or did children just have to answer them all. How was performance scored? More information on how the tests were administered and scored in needed.
  4. On line 251 it is stated “its total score was used as the DV”. Does this refer to the memory for digits task or for the whole battery? How was each subtest scored – I assume one point for each correct answer?
  5. It would be beneficial to remind the reader in the data analysis section (or results) what the factors within the four-factor structure, higher four-factor structure and triple-code are.

Results

  1. Figure 1 appears to be missing the loadings etc.
  2. In section 3.4, please explain at the start of the sentence what test-retest you are interested in.
  3. I would like to see a table that provides descriptive statistics for each of the measures, not just Zareki-R subtests.
  4. Figures depicting each of the three models would be useful so the reader can visually compare across the three.
  5. Line 317: which model are you referring to when you say, “In this model”? This needs to be made clearer.

Discussion

  1. Limitation needs to be added regarding the inclusion criteria (see Method #1).

Minor points

  1. Abstract: “represent” should be “representing”
  2. Remove “in” from line 86
  3. Line 135 to 136: consider writing “as summarised by 45-48” rather than “such as a systematic review…”
  4. Avoid using uncommon acronyms as it increases the working memory load on the reader, which makes it difficult to follow your argument. This comment has been made in response to the use of NC, NP and CA on line 155 onwards.
  5. Line 353, is the word “model” missing?

Author Response

Reviewer #1

Thank you for this careful and extremely useful review. We basically attended all your recommendations. We apologise for the imperfections on the draft. We became aware of this call a month before the deadline, then it was a challenge to provide a cohesive manuscript quickly as authors were from different centres. Finally, we will include reviewers’ support on the Acknowledgements session.

Abstract

  1. I recommend re-reading the abstract as some of the language use is unclear. This includes “accomplishment of elementary”, “based on a battery [of what?]” and “scholar children”. Doing this will enhance clarity of the abstract.

These aspects were addressed.

  1. Specify the low educational attainment country

Not initially mentioned to support anonymity during the submission. But included now since it is an open review.

  1. Be careful with the use of the term “both genders” as society has broadened its understanding of gender and now encompasses terms such as non-binary.

We substituted for sexes. For the record, we absolutely acknowledge and welcome the change! But when the study was carried out, guardians were not asked about non-binary options for their kids. For the record, in our current studies we ask participants to describe gender identity in their own words.

  1. What were the four factors in the best-fit model? This is needed to understand the significant correlations mentioned also.

We agree, it was not included initially to respect the word count.

  1. Can you add a few words to explain what the 1.5 z score is considered useful for?

We meant standard deviation, which is the official clinical diagnosis criterion adopted in this neuropsychological battery (performance -1.5 SD is used for the diagnosis of DD).

  1. Three models are mentioned here (line 20) but then the introduction talks only about two (von Aster and Shalev vs. triple code model).

From von Aster model we tested two models one with only correlated four factors and one hierarchical with 4 factors loading on a higher order factor where the covariation between the four factors was accounted for by a higher-order mathematical cognition (MC) factor. This was explained in the text.

Introduction

The major limitation of the introduction is that it does not adequately explain the models of numerical cognition, the skills that form these models and how they are assessed, and what has been found in relation to both these models in previous literature.

Models were explained, including similarities and differences.

  1. In the first paragraph (line 34), it could be useful to include parentheses offering examples of the “other cognitive abilities”

Included!

  1. On line 43, is this meant to refer to International Classification of Disease rather than International Consensus?

We did a literal citation, including page number, from the consensus article. In this case, the consensus agreed with the medical manuals. Later in the same paragraph and throughout the text we cited the medical manuals.

  1. What is the frequency of secondary DD – this is needed so the reader more fully understands the difference in prevalence rather between primary and secondary forms.

Good point! It makes total sense in a prevalence paper: It is 1 or 2% primary and 4-6% secondary. Authors were cited.

  1. Point three could be better explained (line 56). Do you mean inconsistency between expected school year for chronological age and actual school year? Or is it an inconsistency between expected math performance for their school year and observed math performance?

The response is either. Cognitive deficits traditionally are identified through chronological age discrepancy although there are exceptions. Achievement measures may use both discrepancy criteria for age or/and schooling but more frequently schooling. Note that, regardless of type of instruments (cognitive or achievement measures), most of the studies cited in the paper contrasted grade groups, which is also misleading as the average score of these grade groups might be the result of age scores.

  1. What is considered “high prevalence” (see line 75)?

We meant “higher” than the average prevalence, it was amended.

  1. More information is needed on the difference between low attainers in mathematics and those with severe difficulties given that this is a distinction that is important for this study.

Several examples were included.

  1. For readers outside of Brazil, it would be beneficial to clarify the age of students in year 3 (see line 89). This could also be clarified for all other year groups mentioned.

Currently a 3rd grader is 8 years old. Until 2010 (Law 11.274) a 7-year-old child should be enrolled in the first grade. After that, 1st grade is expected to start at age 6. However, the Law was not automatically in force, schools had a period transitioning to the new rule. Note that we cited in this paper FIVE studies conducted in Brazil some were carried out before this law and some after that, then while comparing different these studies it is challenging to determine if a child was enrolled at the school following the old or new system. It means that the current age/grade relationship (i.e., 8 years old equal 3rd grader) clashes with some references cited which used the previous criterion in which 9 years old were enrolled in the 3rd grade. We added a footnote, without this detailed explanation, to avoid confusion for the readers.

  1. Sentence starting on line 101 and continuing to 104 is unclear. This could be re-worded

It was re-worded (Lines 111-114).

  1. Supporting citations for environmental factors have the stronger effect are needed.

Cited!

  1. Much more information is needed regarding the numeracy related skills that are critical to the development of mathematics. As it stands, after reading the introduction the reader has only a limited understanding of what these skills are.

Two paragraphs refer to the relevant factors and respective subtests. Some subtests are directly inferred from their names, such as “reading numbers”, the specificity of each task, for instance, in this case reading from Arabic form is described on the Method. Therefore, we believe it would be redundant to detail the tasks further on the introduction (Lines 149-177).

  1. Paragraph starting on line 127. In the first line you need to clarify who test you are referring to as this is a new paragraph.

It was included, the Zareki-K (Now, line 162)

  1. Line 135 to 136: consider writing “as summarised by 45-48” rather than “such as a systematic review…”

Done! (Line 157)

  1. Line 149: observed variance in what?

The invariance of the higher-order model across genders (Line 196).

  1. What is important about NC, NP and CA being related to phonological memory?

This was described in von Aster & Shalev (2007), the essential argument being that working memory is important for number line development. The article also describes how working memory and pre-frontal cortex participate along with the intraparietal sulcus in the arithmetic network. Rotzer et al (2009), tested children with DD precisely with digit span and block span and measured through fMRI found correlations between both tasks and the right intraparietal sulcus. As for number processing, see also the classic study by Dehaene et al (2003).

  1. As you are aiming to compare two models of numerical cognition (von Aster and Shalev vs. triple code model) both of these need to be explained in a lot more detail, along with the measures used to assess the aspects of these models. How do they overlap? What has previous research found in relation to each of these models? How do these models relate to your justification for conducting this research in Brazil?

As pointed by Reviewer #2 the introduction is already quite long! But we gave a description of the modules. The models are of general interest, not particularly relevant for Brazil. By contrast, the fact that the study is carried out in a developing country is indeed important for the aim concerned to prevalence because poverty, parent’s low education and precarious educational resources can be confounding factors for the diagnosis. Most prevalence studies were carried out in developed countries.

  1. What compared groups are you referring to here (line 171)? More information is needed.

Age bands (Line 205).

  1. What is the four-factor solution, the higher-four factor solution? These terms have not been used before until the final paragraph of the introduction.

The four factors are Number Sense, Number Comprehension, Number Production and Calculation they were explained in the text. Von Aster & Shalev’s model was tested in two ways, first as the four factors and second as a high order factor. In the higher factor the covariation between the four factors was accounted for by a higher-order mathematical cognition (MC) factor.

  1. The introduction does not provide sufficient detail to support why you are contrasting z scores and percentiles.

We provide core information about the rationale for contrasting z scores and percentiles

with relevant references.

Method

  1. Raven’s coloured progressive matrices assesses non-verbal cognition while the WISC assesses full-scale IQ. These measures are not directly comparable, nor are simply comparing the non-verbal component of the WISC to the RCPM given the differences in the assessments. Additional analysis needs to be conducted to see whether there is a difference in your key findings depending on whether the children met the inclusion criteria using the RCPM or WISC. This is critical as otherwise you are assuming that average intelligence on a non-verbal measure is the same as average intelligence on a more comprehensive measure.

The objective of the study was not to compare intellectual ability measures. In any case, the correlation is established between both measures r=0.67, N=702 (Wilkes & Weigl, 1998). The aim of the task, for the current study as required by CID and DSM was uniquely to differential diagnosis concerning intellectual disability. The average scores and respective standard deviations clearly show that it was not the case. No children were included in this study based exclusively on the IQ measure.

  1. Dictation of numbers: how many digits were in each number? Was there a range (e.g., single digits to triple digits)? More information needing.

Reading numbers had numerals from two to four digits. Tasks in general are not fully detailed due to copyright constraints. The examples presented in the description make explicit the ranges in the respective subtests.

  1. Was there a discontinuation of any of the Zareki-R assessments? Or did children just have to answer them all. How was performance scored? More information on how the tests were administered and scored in needed.

The battery is fully applied to all participants, items may receive 0, 1 or 2 points depending on the task and the quality of the response, being 0 for incorrect and 2 for accurate response without cues or repetitions.

  1. On line 251 it is stated “its total score was used as the DV”. Does this refer to the memory for digits task or for the whole battery? How was each subtest scored – I assume one point for each correct answer?

Zareki-R has 12 subtests, memory for digits in one of them. The extension was from 3 to 6 items (forward order) or 2-5 (backward order), and there were three sequences of each. Performance was scored according to the manual of the Zareki-R. Total score 24 points per order, based on 2 points for each correct sequence and 0 points for incorrect responses. Total: 48 possible points. The score for the memory of digits is not included in the total score as per Zareki-R manual, precisely because it is a working memory rather than numerical cognition subtest. For this reason, all subsequent analyses (i.e., gender and age effects) were calculated for the 11 subtests, total score, and memory for digits separately.

  1. It would be beneficial to remind the reader in the data analysis section (or results) what the factors within the four-factor structure, higher four-factor structure and triple-code are.

Done!

Results

  1. Figure 1 appears to be missing the loadings etc.

We modified the Figure (now figure 2) and included the estimate parameters (loading etc.)

  1. In section 3.4, please explain at the start of the sentence what test-retest you are interested in.

Zareki-R.

  1. I would like to see a table that provides descriptive statistics for each of the measures, not just Zareki-R subtests.

As suggested by the reviewer, we included a table (Table S1) with descriptive statistics for Verbal IQ of Wechsler Intelligence Scale for Children, Raven’s Coloured Progressive Matrices by gender. No gender differences were found.

  1. Figures depicting each of the three models would be useful so the reader can visually compare across the three.

Done!

  1. Line 317: which model are you referring to when you say, “In this model”? This needs to be made clearer.

In the higher-order Mathematical Cognition solution

Discussion

Limitation needs to be added regarding the inclusion criteria (see Method #1).

As pointed on #Method1, both measures were used essentially to detect potential participants with intellectual disability. These measures have cut-off to identify a person with intellectual disability, the ranges are indicated on Participants section. The Wechsler scales clearly distinguish per levels of intellectual disability (e.g., mild, moderate, and severe intellectual disability). RCPM classifies individuals at 5th percentile as intellectually impaired and it can be used with confidence in studies comparing participants with and without intellectual disability (Facon et al 2011. Goharpey et al 2011).

Minor points

  1. Abstract: “represent” should be “representing”
  2. Remove “in” from line 86
  3. Line 135 to 136: consider writing “as summarised by 45-48” rather than “such as a systematic review…”
  4. Avoid using uncommon acronyms as it increases the working memory load on the reader, which makes it difficult to follow your argument. This comment has been made in response to the use of NC, NP and CA on line 155 onwards.
  5. Line 353, is the word “model” missing?

Thank you. There minor points were all accepted.

MANUSCRIPT ATTACHED

Reviewer 2 Report

Santos et al. presented a well written study testing 3 theoretical neurodevelopmental models of numerical cognition based on a widely used battery for mathematical competence assessment (Zareki-R) and compared the prevalence of DD depending on z-scores and percentiles in a large sample (304) of 7-12 years old Brazilian children. The authors did not find any gender difference and identified the Von Aster & Shalev model as the best model. Prevalence of DD varied between 4.6 and 7.4% depending on the criteria (percentiles or z-scores). This is an overall nice and interesting study, which informs both on the psychometric properties of one of the most used batteries to evaluate numerical cognition as well as on the prevalence of DD in Brazil. The sample size is sufficiently large and statistical analyses are correct. Discussions and conclusions are clear. I only have minor comments listed below, mainly concerning the introduction:

  • I found the introduction a bit too long and not very focused. It is not clear until quite late in the introduction what’s the reason for this study and the reader just find well-known information for most of the intro, more appropriate for a didactic type of document than for a research paper. On the other hand the difference between the three tested models (the four-factor structure, the higher four-factor solution and Dehaene's triple code structure) is not clearly explained and have to be inferred from what has been found in previous studies. Also the neurodevelopmental model proposed by von Aster and Shalev that is studied is not well presented while it should given that this is considered the best model in the end. I would rather give more space to better explain these models and cut on non-essential/well-known points such as the fact that differences in prevalence or in individuals defined as DD varies depending on the diagnostic criteria.
  • Page 2 Line 46 “The current classification of primary or secondary DD is based on aetiological elements.” And following. I never heard about this classification (but maybe I just missed it), can the authors please provide a reference for this classification? The words “primary” and “secondary” sound particularly odd, as they seem to imply that DD can be secondary to another deficit. I would have rather used the words “pure DD” and “DD in comorbidity with other neurodevelopmental disorders (such as dyslexia, adhd, etc)”. Domain general non-numerical deficits (such as attention, wm, inhibition etc) are most often present in DD, even in pure DD I think, without for this reason necessarily identifying different subtypes of DD.
  • Page 2 Line 54 “The currently general criteria acknowledged include: i) discrepancy with intelligence measures;”

The psychometric quantification of the discrepancy between the specific skill (e.g., mathematics) and the overall general intelligence is no longer required by DSM-5 (which is quite universally accepted). The authors can refer to Castaldi E, Piazza M, Iuculano T. Learning disabilities: Developmental dyscalculia. Handb Clin Neurol. 2020;174:61-75. doi: 10.1016/B978-0-444-64148-9.00005-3 for an updated overview of the diagnostic criteria as well as of the behavioral and neural characterization of DD and different etiologic accounts.

  • Page 2 “especially in countries with low overall educational attainment countries (24)” typo
  • Page 2 words inverted: both children those who have
  • Page 9 Typo: In order to investigate the neurodevelopmental of numerical cognition we carried out..

Author Response

Reviewer #2

We do appreciate the time you spent contributing to improve our work and your positive views on our study. We did our best to follow all your recommendations. We will include reviewers support on the Acknowledgements session!

I found the introduction a bit too long and not very focused. It is not clear until quite late in the introduction what’s the reason for this study and the reader just find well-known information for most of the intro, more appropriate for a didactic type of document than for a research paper. On the other hand the difference between the three tested models (the four-factor structure, the higher four-factor solution and Dehaene's triple code structure) is not clearly explained and have to be inferred from what has been found in previous studies. Also the neurodevelopmental model proposed by von Aster and Shalev that is studied is not well presented while it should given that this is considered the best model in the end. I would rather give more space to better explain these models and cut on non-essential/well-known points such as the fact that differences in prevalence or in individuals defined as DD varies depending on the diagnostic criteria.

This is a very important point, also made by Reviewer #1. We agree that it is crucial to provide further info about the two core models. We also agree that rationale should appear earlier, and we changed that accordingly. We cannot skip the prevalence debate because one of the aims of the study is precisely to compare two prevalence criteria.

Page 2 Line 46 “The current classification of primary or secondary DD is based on aetiological elements.” And following. I never heard about this classification (but maybe I just missed it), can the authors please provide a reference for this classification? The words “primary” and “secondary” sound particularly odd, as they seem to imply that DD can be secondary to another deficit. I would have rather used the words “pure DD” and “DD in comorbidity with other neurodevelopmental disorders (such as dyslexia, adhd, etc)”. Domain general non-numerical deficits (such as attention, wm, inhibition etc) are most often present in DD, even in pure DD I think, without for this reason necessarily identifying different subtypes of DD.

This definition appears in one of the most cited papers in Developmental Dyscalculia. A paper signed by an international panel composed by fourteen well-known researchers from six countries (Germany, Austria, US, UK, Israel and Switzerland). These terms are also used in epidemiological studies then it is appropriate to adopt them in the present context.

Kaufmann, L.; Mazzocco, M.; Dowker, A.; von Aster, M.; Göbel, S.M.; Grabner, R. H..., Rubinsten, O. Dyscalculia from a developmental and differential perspective. Front Psychol 2013, 4(516). doi: 10.3389/fpsyg.2013.00516

Page 2 Line 54 “The currently general criteria acknowledged include: i) discrepancy with intelligence measures;”

The psychometric quantification of the discrepancy between the specific skill (e.g., mathematics) and the overall general intelligence is no longer required by DSM-5 (which is quite universally accepted). The authors can refer to Castaldi E, Piazza M, Iuculano T. Learning disabilities: Developmental dyscalculia. Handb Clin Neurol. 2020;174:61-75. doi: 10.1016/B978-0-444-64148-9.00005-3 for an updated overview of the diagnostic criteria as well as of the behavioral and neural characterization of DD and different etiologic accounts.

Thanks for suggesting Castaldi et al (2020); we added this reference. We agree that discrepancy with intelligence is widely disputed. However, the criteria D from DSM-5 states (page 67): [“Criterion D. The learning difficulties are not better accounted for by intellectual disabilities, (…). Note: “The four diagnostic criteria (A, B, C and D) are to be met on a clinical synthesis of the individual’s history…”  On Differential Diagnosis (page 73). “The DSM-5 states: Specific learning disorder differs from general learning difficulties associated with intellectual disability because learning difficulties occur in the presence of normal levels of intellectual functioning (i.e., IQ score of at least 70±5)…”]. Therefore, the need for a certain level of intellectual ability is still required for the diagnosis of DD.

Page 2 “especially in countries with low overall educational attainment countries (24)” typo

The repeated word was deleted.

Page 2 words inverted: both children those who have

We rephrased this sentence.

Page 9 Typo: In order to investigate the neurodevelopmental of numerical cognition we carried out.

It was amended.

MANUSCRIPT ATTACHED
